# AdaNav: Adaptive Reasoning with Uncertainty for Vision-Language Navigation

**Xin Ding** [* 1]   **Jianyu Wei** [1]   **Yifan Yang** [2]   **Shiqi Jiang** [2]   **Qianxi Zhang** [2]   **Hao Wu** [3]   **Fucheng Jia** [4]   **Liang Mi** [3]
**Yuxuan Yan** [5]   **Weijun Wang** [6]   **Yunxin Liu** [6]   **Zhibo Chen** [1]   **Ting Cao** [6]

## Abstract

Vision-Language Navigation (VLN) requires agents to follow natural language instructions by grounding them in sequential visual observations over long horizons. Explicit reasoning could enhance temporal consistency and perception–action alignment, but reasoning at fixed steps often leads to suboptimal performance and unnecessary computation. To address this, we propose **AdaNav**, an uncertainty-based adaptive reasoning framework for VLN. At its core is the **Uncertainty-Adaptive Reasoning Block** (UAR), a lightweight plugin that autonomously triggers reasoning. We introduce *Action Entropy* as a policy prior for UAR and progressively refine it through a *Heuristics-to-RL* training method, enabling agents to learn difficulty-aware reasoning policies under the strict data limitations of embodied tasks. Results show that with only *6K* training samples, AdaNav achieves substantial gains over closed-source models trained on *million-scale* data, improving success rate by 20% on R2R valunseen, 11.7% on RxR-CE, and 11.4% in real-world scenes.

## 1. Introduction

As a fundamental capability for embodied agents, Vision-Language Navigation (VLN) requires agents to interpret natural language instructions and continuously ground them in sequential visual observations to execute long-horizon navigation trajectories (Gu et al., 2022; Park & Kim, 2023). Existing VLM-based methods either rely on augmenting navigation with auxiliary modalities (Krantz et al., 2021; Xu et al., 2023; Yin et al., 2024), such as depth, odometry, or

---
[1]University of Science and Technology of China [2]Microsoft Research [3]Nanjing University [4]Central South University [5]Zhejiang University [6]Institute for AI Industry Research (AIR), Tsinghua University. Correspondence to: Zhibo Chen <chenzhibo@ustc.edu.cn>, Ting Cao <tingcao@mail.tsinghua.edu.cn>.

*Proceedings of the $43^{rd}$ International Conference on Machine Learning*, Seoul, South Korea. PMLR 306, 2026. Copyright 2026 by the author(s).

topological maps to strengthen spatial understanding, or instead scale up training on VLN data *without* auxiliary inputs to improve quality and generalization (Zheng et al., 2024; Wei et al., 2025; Yu et al., 2025). However, despite these advances, current methods are still hindered by two major challenges of VLN: (1) Consistent temporal grounding: continuously capturing progress along the trajectory, tracking completed parts, and deciding the next action; (2) Robust perception–action mapping: grounding language in the current spatial context, recognizing landmarks, localizing itself, and selecting appropriate navigation actions.

To address these challenges, explicit reasoning has been introduced to VLN (Zhou et al., 2024b; Wang et al., 2024; Lin et al., 2025a; Chen et al., 2024a), enabling agents to better align language, perception, and action over long-horizon navigation trajectories. However, current straightforward reasoning at each step not only incurs prohibitive computational overhead, but also results in overthinking (Sui et al., 2025; Wu et al., 2025; Shen et al., 2025) that degrades navigation quality (Figure 4 shows higher quality with fewer reasoning steps).

Ideally, VLN agents should exhibit adaptive reasoning, i.e., decide *when and how* to reason. However, achieving such adaptivity and mitigating the overconfidence issue of LLMs (Sun et al., 2025; Groot & Valdenegro-Toro, 2024) typically require large-scale supervised fine-tuning (SFT) with task-specific data (Wen et al., 2024; Lin et al., 2025b). However, embodied interaction data is costly to collect and far from web-scale. Under such limited data conditions, it remains difficult for models to learn when and how to adaptively invoke reasoning. To avoid data limitation, we propose **uncertainty-based adaptive reasoning for vision-language navigation (AdaNav)**, as shown in Figure 1. By defining *Action Entropy* as an indicator of uncertainty, AdaNav uses this as an objective and interpretable heuristic prior to deciding when and how to reason, and then refines this prior gradually through reinforcement learning (RL) to optimize the reasoning trigger policy. By combining the efficiency of heuristic guidance with the optimality of RL, AdaNav does not involve costly labeled reasoning trigger data, but allows the agent to automatically invoke reasoning when necessary to maintain temporal grounding and robust perception–action mapping in long-horizon navi-

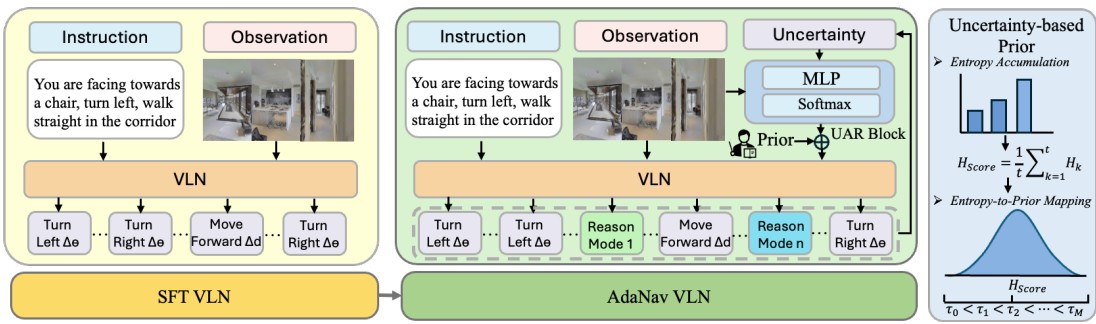

*Figure 1.* AdaNav augments a base VLN model by integrating the Uncertainty-Adaptive Reasoning Block (UAR Block). UAR Block leverages model uncertainty to autonomously trigger reasoning modes and timing, enhancing consistent temporal grounding and perception–action mapping while significantly improving efficiency and mitigating overthinking.

gation. See Figure 2 as an example.

To realize AdaNav, we introduce an *Uncertainty-Adaptive Reasoning Block (UAR Block)* and the *Heuristic-to-RL* training mechanism. UAR block, as a plugin for available VLN models, collects historical, embodied-interaction-dependent uncertainty signals and generates vectorized control signals to dynamically trigger VLN for appropriate reasoning modes. Leveraging the interpretable signals from the UAR Block, the Heuristic-to-RL training first explores the action space under these heuristic priors (e.g., triggering reasoning when uncertainty exceeds a threshold) to guide decision-making at critical moments. As training progresses, the influence of these priors is gradually annealed, allowing RL to autonomously refine the UAR reasoning-trigger policy for optimal reward.

To demonstrate the effectiveness of AdaNav, we integrate it with state-of-the-art open-source VLN backbones and evaluate on classic benchmarks. Remarkably, **with only 6K training samples, AdaNav significantly surpasses closed-source models trained on million-scale data.** Specifically, our method achieves an average 20% improvement in success rate on R2R val-unseen (Krantz et al., 2020), and even without training on the larger and more challenging RxR-CE (Ku et al., 2020), AdaNav yields a 11.7% gain, demonstrating the cross-dataset generality. Additionally, AdaNav exhibits strong robustness in Sim-to-Real deployment, achieving approximately a 11.4% success rate improvement over 150 instructions across four **real-world indoor scenes**. As training proceeds, AdaNav reduces the average number of reasoning steps per trajectory to only **2.5** (over trajectories with an average length of **55 steps**), while the success rate increases 7% compared to reasoning at fixed steps. Notably, 71% of reasoning steps are concentrated on hard trajectories. These results indicate that training makes reasoning more difficulty-aware and mode-adaptive.

## 2. Related Work

**VLN with Auxiliaries.** VLN requires agents to follow free-form linguistic instructions and visual cues to reach a target location. Early studies relied on pre-defined way-points for discrete navigation (Qi et al., 2020b; Thomason et al., 2020) in the Habitat-Matterport3D simulator (Chang et al., 2017), while more recent works (Qi et al., 2020a; An et al., 2021; Hong et al., 2020; Tan et al., 2019; Wang et al., 2019; Shah et al., 2023; Hirose et al., 2024; He et al., 2026) use continuous environments, namely *VLN-CE*, like Habitat (Krantz et al., 2020), enabling low-level actions (e.g., move forward, rotate) for more realistic navigation. With the rise of Transformers, many works introduced pre-trained methods with auxiliary modalities, e.g., depth, odometry, or topological maps, for VLN (Ma et al., 2019; Wang et al., 2019). DUET (Chen et al., 2022) and ETPNav (An et al., 2024) build topological maps for global navigation understanding, while GridMM (Wang et al., 2023) introduced a dynamic egocentric grid memory. Although these methods improve spatial awareness, they inevitably limit generalization and introduce computational overhead and noise (Zhang et al., 2024b). Modern works increasingly target video-only general solutions for VLN without auxiliaries (Zhang et al., 2024b; Cheng et al., 2024; Zhang et al., 2024a). *VLN-CE with only videos captured by the monocular camera is also the target of this paper.*

**Vision-Language Models for Navigation.** With the rapid development of Vision-Language Models (VLMs), RT-2 (Zitkovich et al., 2023) demonstrates the potential of transferring web-scale knowledge from VLMs to generalizable robotic manipulation. Recent work has focused on scaling VLN training data and fine-tuning large VLMs. For example, Navid (Zhang et al., 2024b) used 550k navigation samples to fine-tune Vicuna for navigation; NaV-ILA (Cheng et al., 2024) expanded to 3–5M samples combining real and simulated navigation data plus general VQA supervision; Uni-NaVid (Zhang et al., 2024a) further incorporated 3.6M multi-task trajectories from Habitat-Matterport3D (Chang et al., 2017; Krantz et al., 2020) and real video QA data (Azuma et al., 2022; Ren et al., 2024; Chen et al., 2024b; Li et al., 2024) for cross-task generalization. Despite these advances, VLM-based VLN agents still fall short in task quality, struggling with consistent temporal grounding and robust perception–action mapping, particu-

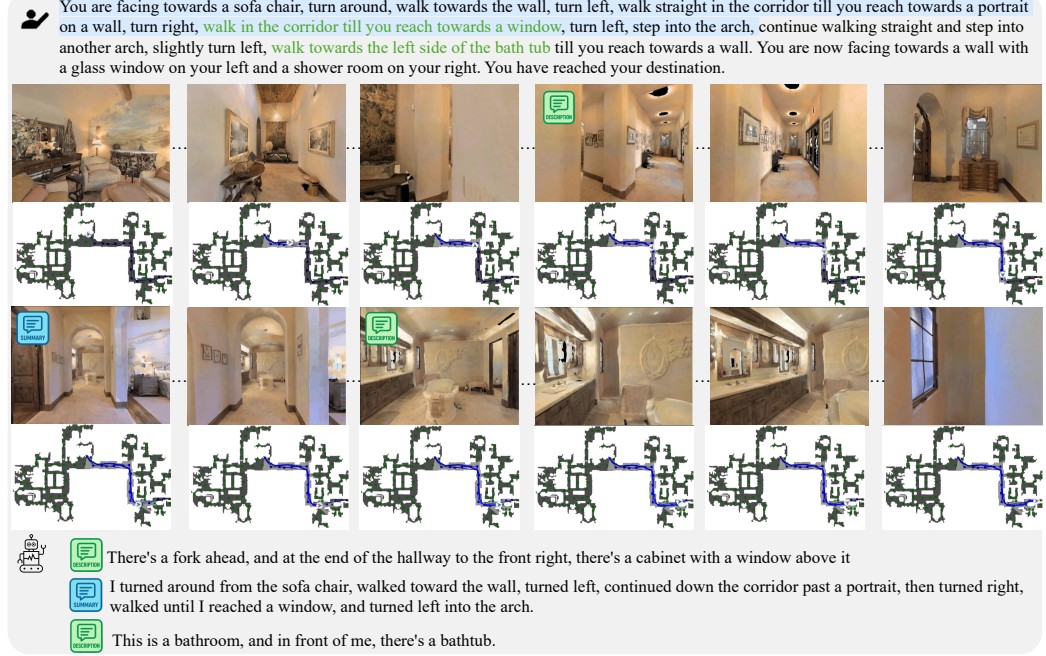

You are facing towards a sofa chair, turn around, walk towards the wall, turn left, walk straight in the corridor till you reach towards a portrait on a wall, turn right, walk in the corridor till you reach towards a window, turn left, step into the arch, continue walking straight and step into another arch, slightly turn left, walk towards the left side of the bath tub till you reach towards a wall. You are now facing towards a wall with a glass window on your left and a shower room on your right. You have reached your destination.

DESCRIPTION There's a fork ahead, and at the end of the hallway to the front right, there's a cabinet with a window above it

SUMMARY I turned around from the sofa chair, walked toward the wall, turned left, continued down the corridor past a portrait, then turned right, walked until I reached a window, and turned left into the arch.

DESCRIPTION This is a bathroom, and in front of me, there's a bathtub.

*Figure 2.* A visualization example of a VLN trajectory executed by AdaNav. AdaNav autonomously invokes reasoning, e.g., summarization (in blue) and description (in green) when necessary to maintain consistent temporal grounding and robust perception–action mapping.

larly in long-horizon trajectories and complex environments.

**Explicit Reasoning for Navigation.** To mitigate these challenges, recent works introduce explicit reasoning via off-the-shelf LLMs, where pre-defined programming rules constrain when and how reasoning modes, e.g., description, summarization, or error correction, are applied. For example, FAST Navigator (Ke et al., 2019) enables self-correction in VLN via backtracking and path re-exploration; LLM-Planner (Song et al., 2023) parses instructions into sub-goals; NavGPT (Zhou et al., 2024b) generates step-wise textual descriptions and historical trajectories; NavGPT-2 (Zhou et al., 2024a) further integrates visual grounding; MiC (Qiao et al., 2023) organizes reasoning into a "summarization–planning–correction" loop; DiscussNav (Long et al., 2024b), MCGPT (Zhan et al., 2024), and Instruct-Nav (Long et al., 2024a) leverage expert collaboration or graphs for error correction and historical summarization.

While these rule-driven frameworks offer interpretability, they inherently restrict flexibility in open-ended environments, hinder efficiency, and may lead to overthinking (Fang et al., 2025; Dai et al., 2025). In contrast, our method will employ a learnable mechanism that enables agents to autonomously decide when and how to reason.

## 3. Method

### 3.1. Problem Formulation of AdaNav

The problem investigated in this work is how to enable an embodied agent to autonomously decide *when* and *how*

to invoke reasoning during VLN. Unlike conventional approaches that either disable reasoning or enforce rule-based reasoning at fixed steps, our goal is to learn an autonomous reasoning policy that dynamically determines the timing and mode of reasoning, optimizing both efficiency and navigation performance.

**Vision-Language Navigation.** We consider a standard VLN setting where an agent is placed in a 3D environment $\mathcal{E}$ with state space $\mathcal{S}$ and action space $\mathcal{A} = \{\text{turn\_left}(\Delta\theta), \text{turn\_right}(\Delta\theta), \text{move\_forward}(\Delta d), \text{stop}\}$, where $\Delta\theta$ and $\Delta d$ denote the angle and distance, respectively. Given a natural language instruction $I$ and sequential visual observations $\{o_1, o_2, \dots\}$, the agent executes a trajectory $\tau = \{(s_t, a_t)\}_{t=1}^{H}$ toward a goal $s^*$ specified implicitly by $I$, aiming to maximize task success:

$$\pi^* = \arg\max_{\pi} \; \mathbb{E}_{\tau \sim \pi}\big[\mathbf{1}(s_H = s^*)\big]. \qquad (1)$$

**Adaptive Reasoning for VLN.** To improve VLN performance in long-horizon and complex environments, we allow explicit reasoning at step $t$ with a *mode* variable $m_t \in \{\varnothing\} \cup \mathcal{M}$ and reasoning content $r_t$. Here, $m_t = \varnothing$ denotes *no reasoning* (so $r_t = \varnothing$), while $m_t \in \mathcal{M}$ denotes invoking a reasoning mode from a predefined set. In this work, we consider three reasoning modes: *description*, *summary*, and *error correction* (see Figure 6 and Appendix G for instances). The agent's policy then consists of two coupled processes: 1) a navigation policy $\pi_{\text{nav}}(a_t \mid h_t^{\text{nav}}, I, r_{\leq t})$, and 2) a reasoning policy $\pi_{\text{rea}}(m_t, r_t \mid h_t^{\text{rea}}, I)$ that jointly decides *when* to reason (via $m_t = \varnothing$ vs. $m_t \neq \varnothing$) and *which*

*mode* to use (via $m_t \in \mathcal{M}$).

The overall joint policy is

$$\pi^*(a_t, m_t, r_t \mid h_t, I) = \pi_{\text{nav}}(a_t \mid h_t^{\text{nav}}, I, r_{\leq t}) \cdot \pi_{\text{rea}}(m_t, r_t \mid h_t^{\text{rea}}, I) \tag{2}$$

where $h_t = (h_t^{\text{nav}}, h_t^{\text{rea}})$ denotes the full history, with $h_t^{\text{nav}}$ and $h_t^{\text{rea}}$ representing the navigation-related and reasoning-related information, respectively. For clarity, we factorize the reasoning policy as:

$$\pi_{\text{rea}}(m_t, r_t \mid h_t^{\text{rea}}, I) = \underbrace{\pi_{\text{txt}}(r_t \mid m_t, h_t^{\text{rea}}, I)}_{\text{reasoning content}} \cdot \underbrace{\pi_{\text{sel}}(m_t \mid h_t^{\text{rea}}, I)}_{\text{when/which mode}} \tag{3}$$

With the constraint $r_t = \varnothing$ if $m_t = \varnothing$. Here, $\pi_{\text{txt}}$ shares the same network as $\pi_{\text{nav}}$.

By integrating navigation and reasoning, the overall learning objective is to jointly optimize both policies, aiming to maximize task performance while maintaining computational efficiency.

$$\pi^* = \arg \max_{(\pi_{\text{rea}}, \pi_{\text{nav}})} \mathbb{E}_{\tau \sim (\pi_{\text{nav}}, \pi_{\text{rea}})} \left[ R_{\text{task}}(\tau) \right] \tag{4}$$

where $R_{\text{task}}(\tau)$ jointly accounts for navigation success (e.g., progress or success indicator) and the latency penalty induced by reasoning calls.

### 3.2. Methodology of AdaNav

**Motivation.** Adaptive reasoning requires the agent to selectively decide *when* reasoning is beneficial and *which mode* to invoke. However, native VLMs are neither sensitive nor objective in perceiving task difficulty, often resulting in overconfidence. In LLM research, similar issues (e.g., in mathematical reasoning) have been alleviated by incorporating high-quality reasoning traces with supervised fine-tuning (Zhang et al., 2025; Tian et al., 2025; Guo et al., 2025). In contrast, for embodied agents, collecting such high-quality interaction traces is prohibitively expensive. This motivates the development of alternative approaches that enable agents to acquire adaptive reasoning capabilities without relying on large-scale reasoning supervision.

To this end, we propose AdaNav, which leverages interpretable uncertainty signals to dynamically trigger reasoning only when necessary. By combining a learnable reasoning policy with a navigation policy, AdaNav enables efficient, difficulty-aware, and mode-adaptive reasoning, achieving high performance in long-horizon and complex VLN tasks.

#### 3.2.1. UNCERTAINTY-ADAPTIVE REASONING BLOCK

Recent works (Kazemnejad et al., 2024; Wang et al., 2025b; Fu et al., 2025) in language reasoning have shown that high-entropy tokens exert a disproportionately large influence on single-step text generation. Inspired by this, we explore whether a similar principle can serve as a signal for identifying "forking steps" in navigation. Specifically, we define

**action entropy** $H$ as an uncertainty measure:

$$H = -\frac{1}{N} \sum_{t=1}^{N} \sum_{v=1}^{V} p_t[v] \log p_t[v], \tag{5}$$

where $N$ is the number of tokens generated, $V$ is the size of the vocabulary, and $p_t[v]$ is the probability of the $v$-th vocabulary token at time step $t$.

To validate the effectiveness of action entropy, we conduct a diagnostic study on navigation trajectories. As shown in Figure 3, episodes with high and sustained entropy are strongly correlated with failures, while successful trajectories maintain consistently low entropy (Different Means). Furthermore, instantaneous entropy alone is insufficient: short-lived spikes do not necessarily imply failure, and many successful trajectories exhibit temporary fluctuations without requiring reasoning (Comparable Extremes).

In contrast, *combining historical action entropy trends with current action entropy states provides a more reliable signal, denoted as $H_{\leq t}$*. Successful trajectories show relatively lower entropy over time, while failure-prone ones accumulate persistently high entropy.

**Method Design.** Motivated by these findings, we design a lightweight Uncertainty-Adaptive Reasoning Block that fuses $H_{\leq t}$ with the current observation $o_t$, forming the reason-related information $h_t^{\text{rea}} = (H_{\leq t}, o_t)$. These signals are combined into a compact control vector:

$$p_{mode}^t = W_1 H_{\leq t} + W_2 o_t + W_3 I + b, \tag{6}$$

which directly parametrizes the reasoning mode logits. From this, the mode selection policy (cf. Equation 3) is given by:

$$\pi_{\text{sel}}(m_t \mid h_t^{\text{rea}}, I) = \text{Softmax}(p_{mode}^t). \tag{7}$$

#### 3.2.2. HEURISTIC-TO-RL TRAINING

Benefiting from the interpretable signals of the UAR Block, we do not require large-scale reasoning annotations. Instead, we propose a Heuristic-to-RL training mechanism, shown in Algorithm 1, that bootstraps policy learning with uncertainty-based heuristics. These priors provide a stable cold-start exploration, enabling the agent to discover useful reasoning patterns without exhaustive supervision. As training progresses, the heuristic influence is gradually annealed, allowing reinforcement learning to refine the reasoning trigger policy. This approach integrates the efficiency of heuristic guidance with the optimality of RL, leading to adaptive long-horizon reasoning strategies that generalize to novel environments.

**Uncertainty-based Prior.** In the cold-start phase, the RL policy has not yet learned meaningful mode selection.

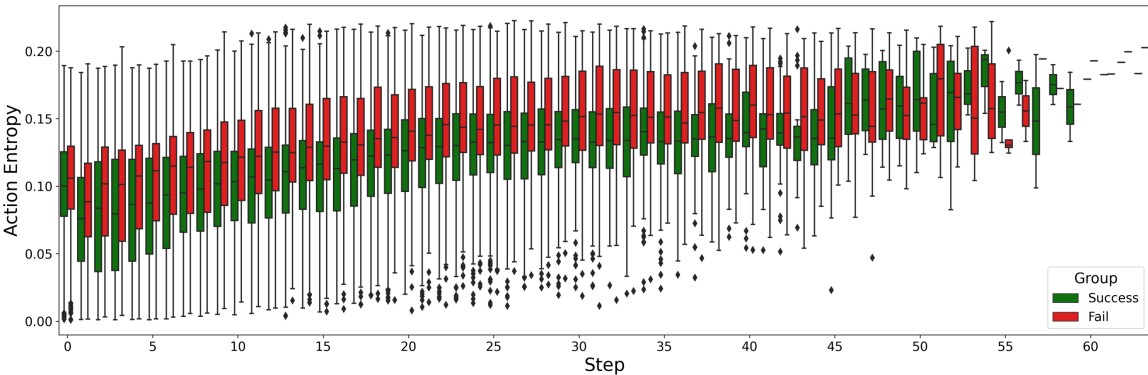

*Figure 3.* Action entropy per step for successful (green) vs. failed (red) trajectories. Failed trajectories show higher, especially in later steps, indicating that policy uncertainty correlates with navigation errors.

We therefore initialize training with an uncertainty-based prior. Intuitively, a higher entropy indicates a higher uncertainty, which requires stronger reasoning. We compute the scalar entropy score as the mean of past entropies, $H_{\text{score}} = \frac{1}{t} \sum_{k=1}^{t} H_k$, and map it into a soft prior distribution over $|\mathcal{M}| + 1$ reasoning modes (including the "no reasoning" option):

$$p_{\text{prior}} = \frac{\exp\left(-|H_{\text{score}} - \tau_m|/\sigma\right)}{\sum_{i=0}^{|\mathcal{M}|} \exp\left(-|H_{\text{score}} - \tau_i|/\sigma\right)}, \quad m = 0, \ldots, |\mathcal{M}| \tag{8}$$

where $\{\tau_0, \tau_1, \ldots, \tau_{|\mathcal{M}|}\}$ are mode-specific entropy thresholds ($\tau_k = \tau_0 + k\delta$), and $\sigma$ controls the smoothness of the prior.

**Heuristic-to-RL Transition.** To gradually shift control from heuristic priors to learned RL policies, we fuse the prior distribution with the model prediction as:

$$p_{\text{final}}^t = \lambda_t \cdot p_{\text{prior}} + (1 - \lambda_t) \cdot p_{\text{mode}}, \tag{9}$$

where $\lambda_t$ is annealed from 1 to 0 over training steps, allowing the RL policy $p_{\text{model}}$ to progressively take over from the uncertainty-based heuristic prior $p_{\text{prior}}$. Accordingly, Equation 7 can be expressed as:

$$\pi_{\text{sel}}(m_t \mid h_t^{\text{rea}}, I) = \text{Softmax}(p_{\text{final}}^t). \tag{10}$$

**Reward Design.** We first define the *reasoning cost* as a normalized penalty based on the relative reasoning length:

$$c_{\text{rea}}(t) = \text{clip}\left(\frac{L_t - L_{\text{shortest\_success}}}{L_{\text{window}}}, 0, 1\right) \tag{11}$$

where $L_i$ is the reasoning length for the current step, $L_{\text{shortest\_success}}$ is the minimal generation length among success samples within the group, and $L_{\text{window}}$ is a constant penalty window.

For the navigation objective, we adopt the common extrinsic reward based on distance reduction, where the immediate reward is defined as $r(s_t, a_t) = D_{\text{target}}(s_t) -$

---

**Algorithm 1** Heuristic-to-RL

1: Initialize navigation policy $\pi_{\text{nav}}$, reasoning selector $\pi_{\text{sel}}$, annealing schedule $\lambda_t$
2: **for** each training episode $\tau$ **do**
3:     **for** each step $t = 1 \ldots T$ **do**
4:         Observe state $o_t$ and entropy $H_{\leq t}$
5:         Compute control vector $p_{\text{mode}}$
6:         Estimate Uncertainty prior $p_{\text{prior}}$
7:         Fuse prior and model (Eq. 9)
8:         Sample reasoning mode $m_t \sim p_{\text{final}}$
9:         **if** $m_t \neq \varnothing$ **then**
10:             Generate reasoning $r_t$
11:         **end if**
12:         Select action $a_t \sim \pi_{\text{nav}}(a_t \mid o_t, r_{\leq t})$
13:         Execute $a_t$, observe next state $s_{t+1}$
14:         Compute extrinsic reward $r(s_t, a_t)$ and reasoning cost $c_{\text{rea}}(t)$
15:     **end for**
16:     Compute task reward (Eq. 12)
17:     Update policy (Eq. 4)
18: **end for**

---

$D_{\text{target}}(s_{t+1}),; t < T$, with $D_{\text{target}}(s_t)$ denoting the geodesic distance from the current state $s_t$ to the target location $s_{\text{target}}$.

Finally, by combining extrinsic reward and reasoning cost, the overall task reward formulated in 4 is defined as the discounted cumulative return:

$$R_{\text{task}}(\tau) = \sum_{t=1}^{T} \beta^{t-1} \left(r(s_t, a_t) - c_{\text{rea}}(t)\right) \tag{12}$$

where $\beta \in (0, 1]$ is the discount factor controlling the weight of future rewards. This formulation encourages the agent to navigate efficiently toward the goal while avoiding unnecessary reasoning overhead.

Overall, this Heuristic-to-RL scheme combines the efficiency of uncertainty-based priors with the optimality of RL, allowing the agent to progressively acquire adaptive reasoning strategies.

*Table 1.* Comparison with the state-of-the-art method on Val-Unseen split of R2R-CE and RxR-CE.

| Method | Observation | | | | R2R-CE Val-Unseen | | | | RxR-CE Val-Unseen | | | | Training Data |
|---|---|---|---|---|---|---|---|---|---|---|---|---|---|
| | S.RGB | Pano. | Depth | Odo. | NE↓ | OS↑ | SR↑ | SPL↑ | NE↓ | OS↑ | SR↑ | nDTW↑ | |
| Open-Source | | | | | | | | | | | | | |
| Seq2Seq | ✓ | | ✓ | | 7.77 | 37.0 | 25.0 | 22.0 | 12.10 | 13.9 | 11.9 | 30.8 | - |
| CMA | ✓ | | ✓ | | 7.37 | 40.0 | 32.0 | 30.0 | - | - | - | - | - |
| RGB-Seq2Seq | ✓ | | | | 10.10 | 8.0 | 0.0 | 0.0 | - | - | - | - | - |
| RGB-CMA | ✓ | | | | 9.55 | 10.0 | 5.0 | 4.0 | - | - | - | - | - |
| LAW | ✓ | | ✓ | ✓ | 6.83 | 44.0 | 35.0 | 31.0 | 10.90 | 8.0 | 8.0 | 38.0 | 150K |
| AO-Planner | | ✓ | ✓ | | 5.55 | 59.0 | 47.0 | 33.0 | 7.06 | 43.3 | 30.5 | 50.1 | 40K (Distill) |
| NaVid | ✓ | | | | 5.47 | 49.0 | 37.0 | 35.0 | 6.79 | 46.2 | 40.5 | 52.2 | 550K |
| Uni-NaVid | ✓ | | | | 5.58 | 53.5 | 47.0 | 42.7 | 6.24 | 48.7 | 40.9 | - | 3600K |
| NaVILA | ✓ | | | | 5.22 | 62.5 | 54.0 | 49.0 | 6.77 | 49.3 | 44.0 | 58.8 | ~3000K |
| Close-Source | | | | | | | | | | | | | |
| NaVid-4D | ✓ | ✓ | | | 5.99 | 55.7 | 43.8 | 37.1 | - | - | - | - | 1840K |
| MonoDream | ✓ | | | | 5.45 | 61.5 | 55.8 | 49.1 | 6.38 | 55.8 | 49.4 | - | 1420K |
| AdaNav | | | | | | | | | | | | | |
| NaVid-AdaNav | ✓ | | | | 5.39 | 57.89 | 47.7 | 42.34 | 6.38 | 58.1 | 47.01 | 56.8 | +6K |
| Uni-Navid-AdaNav | ✓ | | | | 5.11 | 61.6 | 56.9 | 49.5 | 6.20 | 59.6 | 54.5 | 61.5 | +6K |
| NaVILA-AdaNav | ✓ | | | | 5.01 | 66.62 | 60.19 | 50.0 | 6.21 | 60.51 | 49.8 | 62.2 | +6K |

*Table 2.* Cross-dataset performance on the RxR-CE [30] Val-Unseen split, without training on RxR-CE training set. * indicates our reproduction following the original papers.

| Method | RxR-CE Val-Unseen | | | |
|---|---|---|---|---|
| | NE↓ | OS↑ | SR↑ | SPL↑ |
| Open-Source | | | | |
| LAW | 10.87 | 21.0 | 8.0 | 8.0 |
| CM2 | 8.98 | 25.3 | 14.4 | 9.2 |
| Seq2Seq | 11.8 | 5.02 | 3.51 | 3.43 |
| CMA | 11.7 | 10.7 | 4.41 | 2.47 |
| NaVid* | 8.57 | 32.21 | 21.3 | 20.01 |
| NaVILA* | 8.96 | 43.35 | 32.5 | 26.82 |
| Uni-NaVid | 8.08 | 40.9 | 29.5 | 28.1 |
| Close-Source | | | | |
| MonoDream | 8.57 | 35.9 | 25.1 | 21.6 |
| AdaNav | | | | |
| NaVid-AdaNav | 8.21 | 39.21 | 28.95 | 27.73 |
| NaVILA-AdaNav | 8.25 | 48.65 | 38.82 | 31.21 |

*Table 3.* Evaluation of spatial scene understanding performance on the ScanQA Validation split. * and † denote the use of 8 frames and 64 frames, respectively.

| Method | ScanQA Validation | | | | |
|---|---|---|---|---|---|
| | Bleu-4↑ | Rouge↑ | Cider↑ | Meteor↑ | EM↑ |
| 3D Large Multi-modal Models | | | | | |
| 3D-LLM | 7.2 | 32.3 | 59.2 | 12.2 | 20.4 |
| LL3DA | 13.5 | 37.3 | 76.8 | 15.9 | - |
| Chat-3Dv2 | 14.0 | - | 87.6 | - | - |
| Scene-LLM | 12.0 | 40.0 | 80.0 | 16.6 | 27.2 |
| LEO | 13.2 | 49.2 | 101.4 | 20.0 | 24.5 |
| 2D Vision-Langauge-ActionModels | | | | | |
| Uni-NaVid | - | 45.74 | 94.72 | 19.24 | 28.01 |
| NaviLLM | 12.0 | 38.4 | 75.9 | 15.4 | 23.0 |
| NaVILA* | 14.8 | 46.4 | 95.1 | 18.7 | 27.0 |
| NaVILA† | 16.9 | 49.3 | 102.7 | 20.1 | 28.6 |
| NaVILA-AdaNav* | 15.33 | 47.75 | 97.34 | 19.12 | 27.27 |
| NaVILA-AdaNav† | 16.65 | 50.6 | 102.81 | 21.25 | 29.57 |

# 4. Experiments

We conduct experiments to answer the following questions: (1) **Performance Gain**: How much does our proposed AdaNav improve over existing models on VLN-CE benchmarks and general spatial scene understanding tasks? (2) **Reasoning Coordination**: What scheduling strategy has the UAR Block learned, and does it affect navigation efficiency? (3) **Real-World Effectiveness**: How effective is AdaNav when deployed in real-world scenarios?

## 4.1. Performance Gain

**Implementation details. 1. Base models.** AdaNav is designed to be general and can be integrated into existing VLN models with minimal modifications. To demonstrate its strong generalization ability, we adopt two SOTA open-source VLN models, NAVID (Zhang et al., 2024b) and NAV-ILA (Cheng et al., 2024), as our base models. **2. Training setup.** Training is conducted on 4 NVIDIA RTX A100 GPUs. We construct the training set by randomly sampling 3,000 episodes from the training splits of both R2R (Krantz et al., 2020) and RxR (Ku et al., 2020). For rollout collection during training, each episode is rolled out 5 turns, and the learning rate is set to $1 \times 10^{-6}$. **3. Benchmarks.** To assess

both navigation and spatial scene understanding, we evaluate on the val-unseen splits of R2R and RxR for navigation, and on the ScanQA validation set for scene understanding. Detailed settings are provided in Appendix A.

**VLN-CE Benchmarks.** We compare AdaNav with **recent million-scale closed-source models**, including NAVID-4D (Liu et al., 2025), UNI-NAVID (Zhang et al., 2024a), and MONODREAM (Wang et al., 2025a). As shown in Table 1, although closed-source models generally outperform open-source ones, AdaNav achieves substantial gains with only 6K training episodes, improving NAVID and NAVILA by an average of 20% on R2R and 14.6% on RxR, respectively, and surpassing all closed-source baselines.

**Cross-dataset Evaluation.** As shown in Table 2, we test cross-dataset generalization by training AdaNav solely on 3K R2R samples and evaluating zero-shot on RxR Val-Unseen. AdaNav substantially improves base models, surpassing closed-source systems and demonstrating strong transferability.

**Spatial Scene Understanding Benchmarks.** As a general navigation agent, robust spatial scene understanding

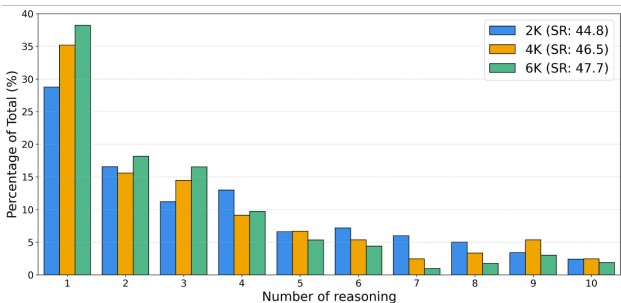

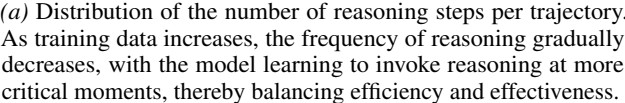

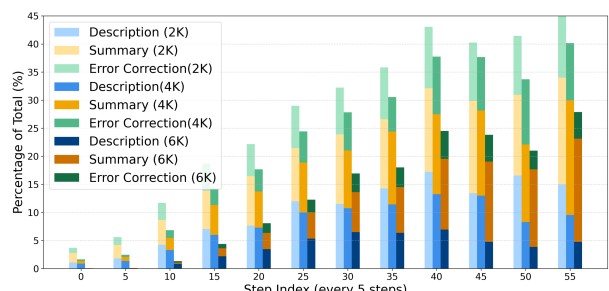

*(a)* Distribution of the number of reasoning steps per trajectory. As training data increases, the frequency of reasoning gradually decreases, with the model learning to invoke reasoning at more critical moments, thereby balancing efficiency and effectiveness.

*(b)* Percentage of reasoning mode invocations at each step, across different training scales. As data increases, reasoning becomes more concentrated on high-uncertainty steps, with stronger preference for *summary* mode at later stages.

*Figure 4.* Analysis of reasoning behaviors in AdaNav across training scales.

(e.g., object localization, referring, and spatial reasoning) is crucial. To verify whether AdaNav fine-tuning affects such capability, we evaluate on the ScanQA validation benchmark (Azuma et al., 2022), a widely used dataset for 3D question answering, as shown in Table 3. Results show that after Heuristic-to-RL training, AdaNav not only preserves its general scene understanding ability without using ScanQA training data, but also achieves slight improvements, indicating enhanced robustness and transferability.

**Real-World Evaluation** To demonstrate the effectiveness of AdaNav in real-time settings, we conduct experiments in real-world environments using 25 simple or complex instructions. Each instruction requires the agent to complete 5–10 sequential landmark-following sub-tasks (e.g., "After passing through the ticket gate, walk straight to the sofa, turn right, take the elevator, continue walking straight, pass through the door until reaching a supermarket, and finally stop at the counter"). Each instruction is executed three times across four types of environments: *Meeting Room*, *Home*, *Office*, and *Outdoor*, following the protocol in prior works (Cheng et al., 2024; Zhang et al., 2024b). The results are summarized in Table 4 and Table 16 in Appendix E.

*Table 4.* Comparing in three diverse real-world environments scenes.

| Method | Meeting Room | | | | Home | | | | Office | | | |
|---|---|---|---|---|---|---|---|---|---|---|---|---|
| | Simple | | Complex | | Simple | | Complex | | Simple | | Complex | |
| | NE↓ | SR↑ | NE↓ | SR↑ | NE↓ | SR↑ | NE↓ | SR↑ | NE↓ | SR↑ | NE↓ | SR↑ |
| Navid | 2.0 | 67.5 | 2.8 | 50.2 | 1.55 | 65.5 | 1.88 | 55.5 | 2.5 | 61.0 | 3.0 | 52.5 |
| Navid-AdaNav | 1.6 | 78.5 | 2.2 | 60.5 | 1.3 | 78 | 1.5 | 75.5 | 2 | 70 | 2.5 | 66.5 |
| Navila | 1.8 | 74 | 2.0 | 65 | 1.0 | 82.5 | 1.4 | 82.5 | 2.1 | 76.8 | 2.2 | 70 |
| Navila-AdaNav | 1.0 | 82.5 | 1.6 | 73.5 | 0.85 | 95 | 1.1 | 88 | 1.5 | 87.6 | 2.1 | 75.5 |

### 4.2. Analysis of UAR Block

To better understand how the UAR Block adapts over training, we conduct a systematic analysis using models trained with 2K, 4K, and 6K data. We focus on two aspects: (1) the frequency and distribution of reasoning invocations across different steps and reasoning modes, and (2) the tendency to trigger reasoning under different episode difficulty levels.

**Frequency of Reasoning** Figure 4a shows the distribution of reasoning steps under different scales of training data, while Table 4a reports the corresponding success rate on R2R. As the training data increases, the model tends to reduce the frequency of reasoning, focusing more on triggering reasoning at critical moments, thereby balancing efficiency and effectiveness.

**Step-wise Reasoning Statistics.** Figure 4b shows the number of reasoning invocations at each navigation step, broken down by mode (*description*, *summary*, *error correction*) for the three training scales. We observe that as training data increases, the agent learns to concentrate reasoning on critical steps where uncertainty is higher, while reducing redundant reasoning in low-uncertainty steps. Additionally, the model increasingly favors *summary* and *error correction* modes at later steps, indicating adaptive mode selection based on task context.

**Difficulty-conditioned Reasoning.** To disentangle how reasoning adapts to task difficulty, we first categorize each episode by its outcome (success vs. failure) under a baseline agent without reasoning coordination. We treat successful episodes as relatively easy and failed episodes as harder ones. We then re-run the model with the coordination layer enabled and analyze reasoning triggers across these two difficulty groups.

*Table 5.* Proportion of reasoning triggers in hard episodes (success + failure = 100%, excl. Step 0). Agents tend to invoke more reasoning in harder episodes.

| Step | 0 | 5 | 10 | 15 | 20 | 25 | 30 | 35 | 40 | 45 | 50 | 55 |
|---|---|---|---|---|---|---|---|---|---|---|---|---|
| failure | 0 | 100.0 | 85.0 | 71.43 | 74.51 | 70.43 | 69.81 | 77.33 | 80.30 | 72.55 | 75.0 | 75.68 |

As shown in Table 5, for hard episodes that the base model fails to solve, reasoning is triggered significantly more frequently. This indicates that the UAR Block adaptively allocates reasoning capacity, focusing on challenging scenarios rather than applying reasoning uniformly across all tasks.

**Conclusions.** These analyses demonstrate that the UAR Block effectively learns both *when* and *which mode* to reason. As training progresses, reasoning becomes more temporally focused, mode-adaptive, and difficulty-aware, enabling the agent to improve navigation performance while minimizing redundant reasoning overhead.

## 5. Ablation

To examine the robustness of AdaNav and assess whether its performance is overly dependent on specific hyperparameter choices, we conduct a series of ablation studies. Our analyses focus on two aspects: (1) Component ablation, (2) Efficiency and Effectiveness of the UAR Block. More detailed ablation results are provided in Appendix B.

**Component Ablation.** We use Navid as the base model and remove or replace major components to isolate their contributions. **(i) w/o UAR Block:** reasoning is invoked at a fixed step (5 step) interval or randomly, without adaptive control. **(ii) w/o Heuristic Prior:** the agent relies purely on reinforcement learning from scratch without uncertainty-based heuristic. **(iii) w/o RL Fine-tuning:** reasoning triggers are guided only by heuristic signals without further policy refinement. Results show that removing either coordination or RL fine-tuning leads to significant performance degradation, confirming that both adaptive gating and learned refinement are essential.

*Table 6.* Component ablation on R2R-CE Val-Unseen. Here, * denotes fixed-interval (5 steps) triggering, and † denotes random triggering..

| Component | NE↓ | OS↑ | SR↑ | SPL↑ |
|---|---|---|---|---|
| Navid | 5.47 | 49.0 | 37.0 | 35.0 |
| w/o UAR* | 5.45 | 53.25 | 40.12 | 38.83 |
| w/o UAR† | 5.44 | 52.10 | 39.5 | 38.65 |
| w/o RL | 5.44 | 55.33 | 42.53 | 39.65 |
| w/o HP | 5.41 | 55.73 | 43.82 | 40.12 |
| AdaNav | 5.39 | 57.89 | 47.7 | 42.34 |

**Computational Efficiency of the UAR Block** The UAR Block adds negligible latency while substantially improving navigation performance. To quantify its computational cost, we measure the latency of each component in Table 7.

*Table 7.* Latency of UAR and policy stages.( P90 Latency represents the latency threshold satisfied by 90% of samples.)

| Component | Peak Latency | Avg.Latency | P90 Latency |
|---|---|---|---|
| UAR Block | 0.0039 s | 0.0029 s | 0.0037 s |
| Base Action Policy | 1.31 s | 0.76 s | 1.00 s |
| Text Reasoning | 5.01 s | 2.94 s | 3.81 s |

As shown in Table 7, the UAR Block introduces **negligible additional computation**, accounting for less than 0.4% of the backbone inference time. Although a single text reasoning step is approximately 3.9× slower than the base action policy inference, tthe adaptive reasoning mechanism

learned by the UAR Block effectively limits the number of reasoning steps to only 2.5 per trajectory on average (with an average trajectory length of 55 steps), resulting in a **minimal overall computational overhead**.

**Effectiveness of the UAR Block Design** As discussed in Section 3.2.1, *action entropy* serves as a reliable statistical indicator of model uncertainty, offering an interpretable and stable heuristic prior for reinforcement learning. At the same time, contextual features extracted from visual observations and language instructions remain essential, as they allow the policy to ground uncertainty in both the physical environment and the intended task objective.

*Table 8.* Ablation on the Input of the UAR Block

| Component | NE↓ | OS↑ | SR↑ | SPL↑ | Avg.Rea |
|---|---|---|---|---|---|
| Navid | 5.47 | 49.0 | 37.0 | 35.0 | 0 |
| Only Action Entropy | 5.41 | 56.73 | 45.23 | 41.55 | 4.5 |
| Only Observation | 5.43 | 55.23 | 43.12 | 40.74 | 8.7 |
| Navid-AdaNav | 5.39 | 57.89 | 47.7 | 42.34 | 2.5 |

Through Heuristic-to-RL Training, AdaNav progressively aligns three key elements—(i) environmental interaction dynamics captured by observations, (ii) internal uncertainty signals captured by action entropy, and (iii) task intent from instructions—ultimately enabling the UAR Block to learn a coherent and difficulty-aware reasoning trigger policy.

As shown in Table 8, when trained with the same 6K samples, models relying solely on action entropy or observation features exhibit a 2–4× increase in reasoning frequency, while suffering a 5–10% drop in navigation success rate. These results indicate that combining contextual features with uncertainty signals is essential for efficient reasoning and high-performance navigation.

## 6. Conclusion

In this work, we tackled the long-standing challenges of consistent temporal grounding and robust perception–action mapping in Vision-Language Navigation. We proposed AdaNav, an uncertainty-based adaptive reasoning framework that integrates the UAR Block with a Heuristic-to-RL training mechanism. This design enables agents to invoke reasoning adaptively, guided first by interpretable heuristic priors and then refined through reinforcement learning, without relying on costly labeled reasoning data. Extensive experiments show that AdaNav delivers substantial improvements: surpassing million-scale closed-source models with only 6K samples, generalizing effectively across R2R and RxR, and demonstrating strong robustness in real-world deployment. Moreover, AdaNav reduces reasoning frequency while making it more difficulty-aware and mode-adaptive, striking a balance between efficiency and effectiveness. AdaNav provides a principled and practical step toward scalable, adaptive reasoning in embodied agents.

## Acknowledgements

This work was supported in part by the Tsinghua University (AIR)-AsiaInfo Technologies (China), Inc. Joint Research Center for 6G Network and Intelligent Computing under Grant 20233910006, the Wuxi Research Institute of Applied Technologies, Tsinghua University under Grant 20242001120, the National Natural Science Foundation of China (NSFC) under Grants 62371434, U25B2010, 62432004, and 62302207, and the Fundamental Research Funds for the Central Universities under Grant 2026300278.

## Impact Statement

This paper presents work whose goal is to advance the field of Machine Learning. There are many potential societal consequences of our work, none which we feel must be specifically highlighted here.

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

# A. Implementation Details

**Real-World Evaluation**   We provide a detailed description of our real-robot platform in Figure 5. The mobile robot is equipped with basic locomotion capabilities and augmented with a camera, microphone, speaker, and LiDAR sensors for user interaction and environment perception. Notably, our method relies only on RGB images and does not require LiDAR input. The system is powered by a Jetson AGX Orin (nvi) running Ubuntu 24.04 with ROS2 Jazzy. In addition, the platform integrates a 19V power regulator and a 110V/220V inverter (both rated at 500W+) to support the compute and sensor modules.

Building on this platform, we design a pipeline for vision-and-language navigation with AdaNav. We experiment with AdaNav on two base models, NAVID (Zhang et al., 2024b) and NAV-ILA (Cheng et al., 2024). In deployment, AdaNav runs on a server with a Jetson that receives compressed images from the robot and sends back high-level commands. The robot then executes these commands (e.g., "Turn right" or "Move forward") through its locomotion system. During navigation, the robot continuously monitors its motion to ensure that rotations and forward movements remain aligned with the issued commands.

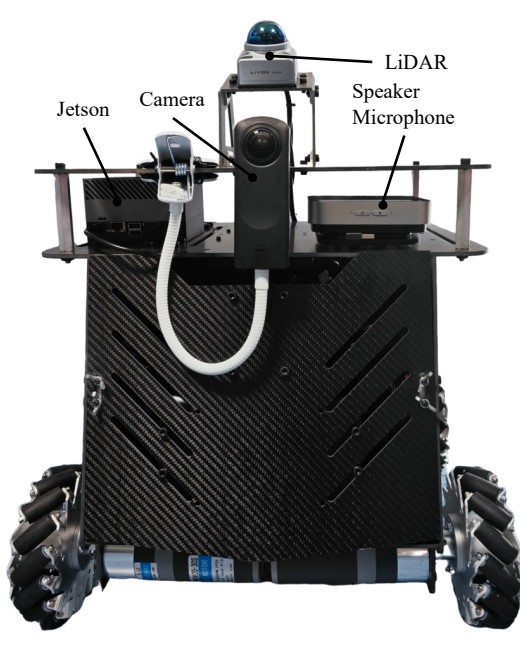

*Figure 5.* Hardware setup of the mobile robot platform used for real-world evaluation.

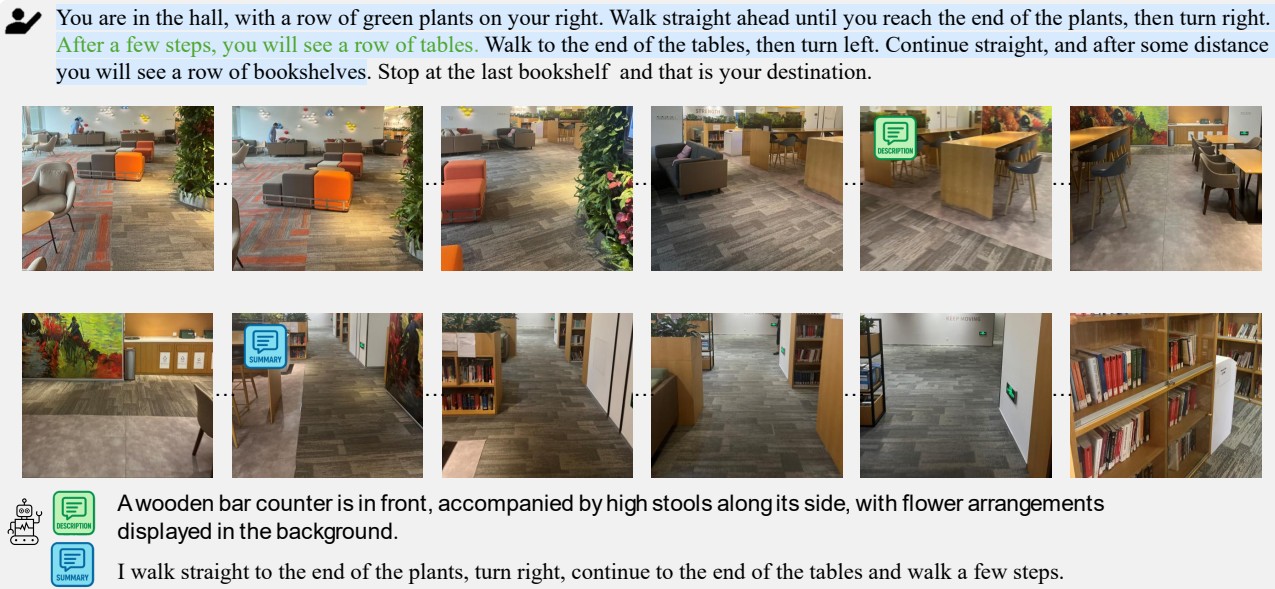

*Figure 6.* Visualization of adaptive reasoning navigation in real-world scene.

# B. Detailed
# Ablation Analysis for Robustness and Sensitivity

**The Impact of Reasoning Mode**   To further analyze our method, we conduct ablations on different reasoning modes. As shown in Figure 4b, different modes contribute at different stages of navigation. This raises the question: *how much would performance degrade if only a single reasoning mode were available?* To investigate, we construct variants where only one reasoning mode is enabled (i.e., description, summary, or error correction). In addition, we add a control setting, where the model is simply instructed with a generic prompt such as: *"At this point, perform some analysis based on the past*

*trajectory."* For all these variants, we apply UAR Block together with the heuristic-to-RL training mechanism, and report the results as follows.

*Table 9.* Ablation results under different *reasoning modes*, where ①: Description only, ②: Summary only, ③: Error correction only, and ④: Generic prompt reasoning. All variants are trained with UAR Block and the heuristic-to-RL mechanism.

| Method | R2R-CE Val-Unseen | | | |
|---|---|---|---|---|
| | NE↓ | OS↑ | SR↑ | SPL↑ |
| AdaNav | 5.39 | 57.89 | 47.7 | 42.34 |
| AdaNav ① | 5.41 | 55.34 | 44.52 | 40.88 |
| AdaNav ② | 5.42 | 56.53 | 46.01 | 41.37 |
| AdaNav ③ | 5.40 | 56.12 | 45.77 | 41.67 |
| AdaNav ④ | 5.40 | 55.23 | 45.62 | 42.01 |

**Comparison with Fixed-Interval Reasoning Strategies**   To investigate whether adaptive uncertain inference can result in a higher actual reasoning frequency while maintaining better performance, we additionally evaluated a series of fixed-interval reasoning strategies (1, 3, 5, 7, 9, 15 steps) without RL training.

*Table 10.* Navigation performance of Navid and fixed-interval reasoning strategies without RL training.

| Component | NE↓ | OS↑ | SR↑ | SPL↑ |
|---|---|---|---|---|
| Navid | 5.47 | 49.0 | 37.0 | 35.0 |
| fixed 1-step | 5.49 | 51.2 | 38.2 | 36.5 |
| fixed 3-step | 5.47 | 50.35 | 39.51 | 37.87 |
| fixed 5-step | 5.45 | 53.25 | 40.12 | 38.83 |
| fixed 7-step | 5.45 | 52.56 | 40.1 | 37.55 |
| fixed 9-step | 5.43 | 51.25 | 40.02 | 38.35 |
| fixed 15-step | 5.45 | 52.23 | 39.2 | 37.48 |
| Navid-AdaNav | 5.39 | 57.89 | 47.7 | 42.34 |

As shown in Table 10, Navid-AdaNav consistently outperforms all fixed-interval strategies, demonstrating that the RL-based UAR Block enables more effective reasoning, achieving both higher navigation performance and efficient adaptive inference.

**Hyperparameter Sensitivity.**   The key hyperparameters in our framework lie in the *Heuristic-to-RL* stage, where we introduce mode-specific entropy thresholds: $(\tau_0, \delta)$ that govern reasoning triggers prior, and a scaling factor $\sigma$.

As shown in Table 6, a well reasoning prior significantly facilitates training. Specifically, $\tau_0$ is estimated from the base model by analyzing 1,000 validation episodes and selecting a percentile of the step-wise action entropy extrema. We experiment with percentiles at 80%, 85%, and 90%, which define progressively stricter confidence thresholds. On top of this, $\delta$ incrementally shifts the thresholds for higher reasoning modes, thereby shaping the curriculum schedule. The corresponding results are summarized in Table 11a and Table 11b. **The results indicate that AdaNav exhibits strong robustness to hyperparameter choices and is not sensitive to hyperparameter variations.**

*Table 11.* Ablation on hyperparameter sensitivity on R2R-CE Val-Unseen.

| $\tau_0$ | $\delta$ | NE↓ | OS↑ | SR↑ | SPL↑ |
|---|---|---|---|---|---|
| | 0.1 | 5.43 | 57.72 | 47.53 | 42.28 |
| 80% | 0.2 | 5.42 | 57.92 | 47.65 | 41.89 |
| | 0.3 | 5.42 | 58.01 | 47.22 | 42.25 |
| | 0.1 | 5.40 | 58.75 | 47.52 | 43.87 |
| 85% | 0.2 | 5.39 | 57.89 | 47.7 | 42.34 |
| | 0.3 | 5.39 | 58.81 | 47.34 | 43.85 |
| | 0.1 | 5.47 | 57.98 | 47.83 | 42.12 |
| 90% | 0.2 | 5.48 | 57.80 | 47.26 | 42.14 |
| | 0.3 | 5.43 | 57.78 | 48.15 | 43.23 |

*(a) Effect of $(\tau_0, \delta)$.*

| $\sigma$ | NE↓ | OS↑ | SR↑ | SPL↑ |
|---|---|---|---|---|
| 0.05 | 5.43 | 57.53 | 47.14 | 42.15 |
| 0.15 | 5.39 | 57.89 | 47.7 | 42.34 |
| 0.25 | 5.44 | 57.51 | 46.78 | 42.55 |
| 0.30 | 5.48 | 57.55 | 47.6 | 41.76 |

*(b) Effect of $\sigma$.*

# C. Additional Experiments with the Uni-NaVid Backbone

To further evaluate the effectiveness of AdaNav, we conducted additional experiments using the newly released Uni-NaVid backbone. We also extended the evaluation to additional continuous-control tasks, including:

- Object Goal Navigation on Matterport3D, and

- Object Goal Navigation on HM3D-OVON.

The results of these experiments are summarized in Tables 12, 13, and 14.

*Table 12.* Results on Object Goal Navigation (Matterport3D and HM3D-OVON).

| Method | SR | SPL |
|---|---|---|
| PIRLNav-IL-RL | 70.4 | 34.1 |
| OVRL | 62.0 | 26.8 |
| OVRL-v2 | 64.7 | 28.1 |
| Uni-NaVid | 73.7 | 37.1 |
| **Uni-NaVid-AdaNav** | **80.15** | **42.2** |

*Table 13.* Results on OVON tasks (Val Seen, Seen Synonyms, Val Unseen).

| Method | SR | SPL | SR | SPL | SR | SPL |
|---|---|---|---|---|---|---|
| | Val Seen | | Seen Synonyms | | Val Unseen | |
| DAgRL | 41.3 | 21.2 | 29.4 | 14.4 | 18.3 | 7.9 |
| VLFM | 35.2 | 18.6 | 32.4 | 17.3 | 35.2 | 19.6 |
| DAgRL+OD | 38.5 | 21.1 | 39.0 | 21.4 | 37.1 | 19.8 |
| Uni-NaVid | 41.3 | 21.1 | 43.9 | 21.8 | 39.5 | 19.8 |
| **Uni-NaVid-AdaNav** | **50.62** | **25.62** | **52.61** | **27.8** | **47.8** | **26.2** |

*Table 14.* Results on R2R and RxR benchmarks.

| Method | R2R | | | | | RxR | | | | |
|---|---|---|---|---|---|---|---|---|---|---|
| | TL | NE↓ | OS↑ | SR↑ | SPL↑ | TL | NE↓ | OS↑ | SR↑ | SPL↑ |
| Uni-NaVid | 9.71 | 5.58 | 53.3 | 47.0 | 42.7 | 15.8 | 6.24 | 55.5 | 48.7 | 40.9 |
| **Uni-NaVid-AdaNav** | **9.91** | **5.11** | **61.6** | **56.9** | **49.5** | **17.7** | **6.20** | **59.6** | **54.5** | **44.2** |

These additional results demonstrate that AdaNav consistently improves navigation performance across different backbones, environments, and task settings, including the newly released Uni-NaVid backbone.

# D. Comparison of Alternative Uncertainty Measures

To evaluate different uncertainty signals for guiding adaptive reasoning, we designed and compared several measures: **value uncertainty**, **value disagreement**, and **temporal variance**, in addition to **action entropy**, which serves as our core uncertainty signal. All signals were trained using the Heuristics-to-RL training method proposed in our work.

### D.1. Value Uncertainty

Following prior work (Wang et al., 2019), an additional critic head is trained to estimate the expected return. The variance across multiple value heads

$$\sigma_V^2 = \text{Var}(V_t^{(i)})$$

provides a measure of value-based uncertainty, capturing inconsistencies in future reward estimation.

## D.2. Value Disagreement

Value disagreement quantifies diversity among predictions of multiple critic heads. Specifically, the training samples are split into three subsets, and one critic head is trained on each subset. The disagreement is computed as the average pairwise squared difference:

$$D_V = \frac{2}{N(N-1)} \sum_{i<j} (V_t^{(i)} - V_t^{(j)})^2,$$

where $V_t^{(i)}$ denotes the value predicted by the $i$-th head at time $t$. Larger $D_V$ indicates greater disagreement among the value heads.

## D.3. Temporal Variance

Temporal variance measures short-term fluctuations of action entropy over a sliding window of length $w$:

$$\sigma_H^2 = \frac{1}{w} \sum_{k=t-w+1}^{t} (H(\pi_k) - \bar{H})^2, \quad \bar{H} = \frac{1}{w} \sum_{k=t-w+1}^{t} H(\pi_k),$$

where $H(\pi_k)$ is the entropy of the action distribution at step $k$. Large $\sigma_H^2$ indicates unstable decision confidence.

Table 15. Comparison of alternative uncertainty measures on navigation performance.

| Component | NE↓ | OS↑ | SR↑ | SPL↑ | Avg. reasoning num |
|---|---|---|---|---|---|
| NaVid | 5.47 | 49.0 | 37.0 | 35.0 | 0 |
| Value Uncertainty | 5.41 | 54.31 | 44.69 | 40.67 | 4.8 |
| Value Disagreement | 5.44 | 54.78 | 43.73 | 40.53 | 5.5 |
| Temporal Variance | 5.40 | 54.65 | 45.21 | 41.15 | 4.2 |
| NaVid-AdaNav | **5.39** | **57.89** | **47.7** | **42.34** | **2.5** |

As shown in Table 15, using action entropy as the core uncertainty signal provides better perception of task difficulty, achieves higher navigation performance, and improves reasoning efficiency compared with alternative uncertainty measures.

# E. Preliminary Exploration in Outdoor VLN Tasks

Current VLN base models perform poorly on outdoor tasks due to several challenges. First, training data for outdoor VLN is scarce, as most available datasets focus on indoor navigation. Second, generalization from indoor to outdoor environments is limited, even with existing indoor datasets. As a result, current models achieve low performance on outdoor VLN benchmarks.

To investigate this scenario, as shown in Table 16, we conducted experiments on the **OpenUAV (Schmittle et al., 2018) outdoor VLN benchmark** as well as in **real-world environments**, including both simple and complex settings. Remarkably, even without access to outdoor training data, **AdaNav** was able to improve the performance of the base model. This improvement is attributed to AdaNav's reasoning mechanism, which leverages knowledge acquired by the base VLM from large-scale internet datasets during pretraining, partially compensating for the scarcity of outdoor data during post-training.

Table 16. Performance of VLN models on OpenUAV and real-world outdoor tasks.

| | OpenUAV | | | | Real World-Simple | | Real World-Complex | |
|---|---|---|---|---|---|---|---|---|
| | NE↓ | SR↑ | OSR↑ | SPL↑ | SR↑ | NE↓ | SR↑ | NE↓ |
| CMA | 102.92 | 14.83 | 22.49 | 13.90 | - | - | - | - |
| Navila | 93 | 21.5 | 45.1 | 21.5 | 55.0 | 2.5 | 35.0 | 3.5 |
| AdaNav | 85 | 25.5 | 50.1 | 23.8 | 67.0 | 1.9 | 45.0 | 2.5 |

## F. Analysis of UAR Block Failure Cases and Improvement

The UAR Block occasionally fails to trigger reasoning under transient ambiguities, such as vague instructions or sudden changes in lighting and viewpoint. In these cases, the action entropy may fluctuate briefly but does not remain high enough to surpass the reasoning threshold, resulting in missed reasoning opportunities.

To address this issue, we conducted RL fine-tuning on data specifically designed to cover such scenarios. During this process, the model learns to associate transient uncertainty with the need for reasoning. We collected 500 samples exhibiting instruction vagueness and lighting changes and conducted additional evaluations on R2R and RxR benchmarks,as shown in Table 17.

*Table 17.* Performance comparison of UAR Block on R2R and RxR benchmarks after targeted RL fine-tuning.

| Method | R2R | | | | RxR | | | |
|---|---|---|---|---|---|---|---|---|
| | NE↓ | OS↑ | SR↑ | SPL↑ | NE↓ | OS↑ | SR↑ | nDTW↑ |
| NaVid | 5.47 | 49.0 | 37.0 | 35.0 | 6.79 | 46.2 | 40.5 | 52.2 |
| NaVid-AdaNav | 5.39 | 57.89 | 47.7 | 42.34 | 6.38 | 58.1 | 47.01 | 56.8 |
| NaVid-AdaNav* | 5.35 | 58.72 | 48.85 | 42.88 | 6.12 | 59.62 | 48.55 | 57.2 |

These results indicate that tailoring the training environment to address transient ambiguity scenarios further improves navigation performance and reduces UAR Block failure cases.

## G. Visualization Results on Adaptive Reasoning Navigation

To better understand how AdaNav adaptively allocates reasoning during navigation, we visualize example trajectories in Figure 7. The figure illustrates both the agent's path and the steps where reasoning is invoked. As shown, AdaNav selectively triggers reasoning at critical or challenging moments, while skipping unnecessary steps in simpler regions of the environment. This demonstrates that the Uncertainty-Adaptive Reasoning (UAR) Block effectively guides the agent to balance efficiency and accuracy.

Figure 7 also highlights that reasoning is concentrated on hard trajectories: compared to the first, simpler scenario, the second, more complex instruction involves more reasoning steps. This observation is consistent with our quantitative analysis in Table 5. These visualizations confirm that the agent's reasoning behavior is both difficulty-aware and mode-adaptive, providing interpretability and insight into its decision-making process.

Your few steps away the entrance, facing towards the pillar, turn slightly left and go straight near the another door, towards your left, enter in to the room, walk through the walkway and enter in to the corridor, from there turn left and go near the door which is open, now you are at the entrance of the room, in front of you there is a dinning table, this would be your final destination.

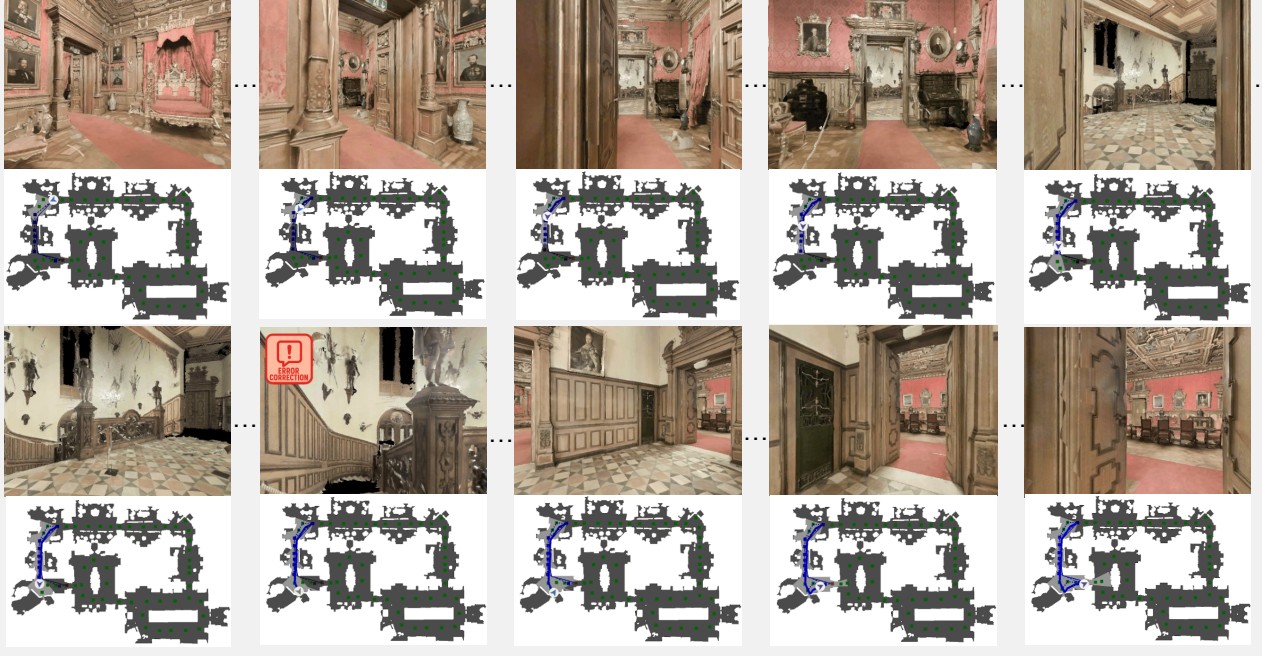

I started near the entrance facing a pillar, then veered slightly left and headed toward a nearby door on my left. After entering the room, I moved along the path inside and made my way into the corridor. I have now reached a staircase landing, which does not match the given instructions

You're facing towards the wooden cabinet onto your right side, now slightly turn to your left, and walk a few steps forward, you can see an ash color sofa onto your right and from there turn to your left, walk a few steps forward and towards your right you can see a wooden table with a plant on it. Walk a few steps forward, you can see an open entrance in front of you, now from the open entrance, turn to your right you can see an open wooden door, enter the door, walk a few steps forward and towards your left you can see another open door, enter it. Now you're in a bedroom, go straight and from the bed, turn towards your left, you enter into a washroom, you're facing towards the sink and that is your destination.

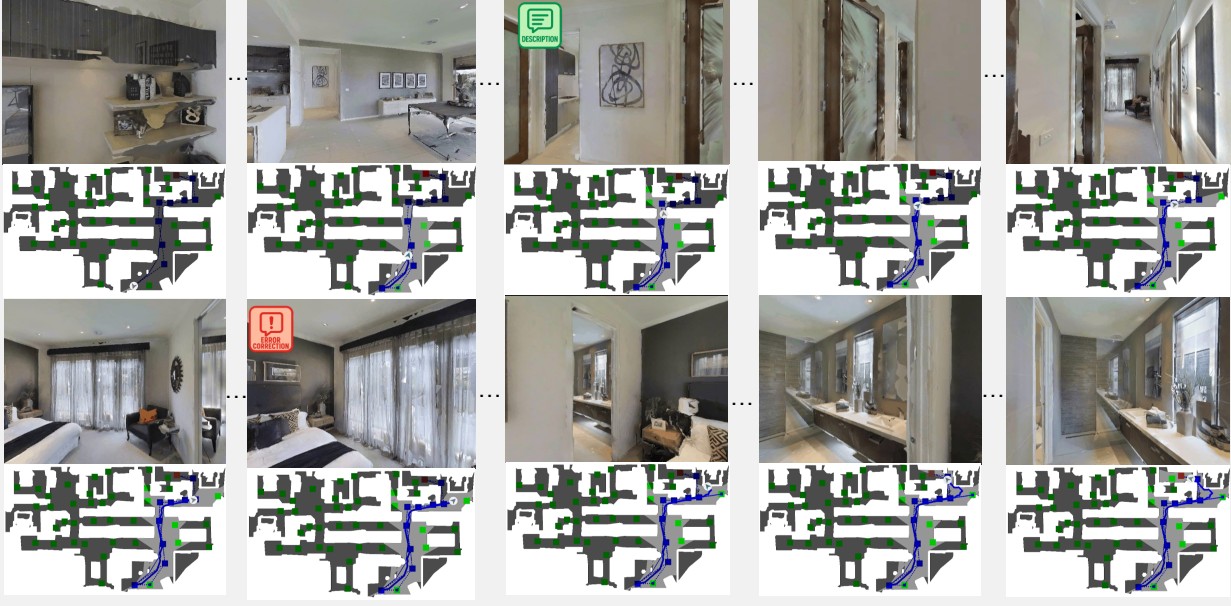

A painting is hanging in the center of the wall. On the left side, there is a doorway leading to a kitchen area with dark cabinets. Wooden-framed glass doors are visible on both the left and right sides.

I moved through the living area, passed a sofa and a wooden table, entered successive doors, and finally arrived in the bedroom, which match the given instructions.

*Figure 7.* Visualizations of adaptive reasoning navigation, where AdaNav autonomously invokes reasoning at high-uncertainty points to