# OpenReview forum: "AdaNav: Adaptive Reasoning with Uncertainty for Vision-Language Navigation"
_ICML.cc/2026/Conference — ICML 2026 regular_

### Official Review · Reviewer_aWMz · 2026-03-11

**Soundness:** 3
**Presentation:** 3
**Significance:** 3
**Originality:** 3
**Overall Recommendation:** 5
**Confidence:** 4

**Summary:**

This is the first navigation framework that leverages uncertainty signals to guide adaptive reasoning. By introducing an Uncertainty-Adaptive Reasoning (UAR) Block and a Heuristic-to-RL training paradigm, the model dynamically triggers reasoning during navigation when necessary, effectively addressing the suboptimal performance and computational overhead caused by fixed-step reasoning.

**Compliance With Llm Reviewing Policy:**

Affirmed.

**Final Justification:**

Since my concerns have been fully resolved, I have decided to keep my positive rating.

**Key Questions For Authors:**

1. In Equation 12’s total reward function, how is the relative weight between reasoning cost and navigation reward determined? If the reasoning penalty is significantly increased or decreased, how would the distribution of reasoning steps generated by AdaNav (Figure 4a) shift? Could adjusting this weight enable flexible switching between a high-performance mode and a high-efficiency (low-cost) mode?
2. At critical moments that require historical summarization, would performance degrade if the model mistakenly invokes environment description instead? And if the selection among the three modes were random, how large would the performance difference be?
3. In Section 3.1 and Figure 6, "error correction" is introduced as one of the reasoning modes. Does the text output directly force the base navigation policy to alter its next action distribution, or is there an explicit rule-based override? How is the agent guaranteed to correct its path rather than just verbally acknowledging an error?
4. In Table 9 (Appendix B), you demonstrate that models restricted to a single reasoning mode (e.g., only "Description" or only "Summary") perform worse than the full AdaNav. However, AdaNav(Summary only) and AdaNav(Error correction only) still achieve surprisingly high performance (OS around 56%) compared to the base NaVid (OS 49.0%). Does this imply that the act of pausing and generating any context-aware text is doing most of the heavy lifting, rather than the specific dynamic switching between the three modes?

**Limitations:**

Please give a discussion on the limitations of this work.

**Strengths And Weaknesses:**

Strengths:

1.This paper proposes a well-motivated approach to the important problem of selective reasoning in (VLN). It introduces a clear formulation that leverages uncertainty signals to guide adaptive reasoning, providing a rigorous and principled design. The ablation studies are thorough and effectively validate the contributions of the AdaNav framework.
2.The data efficiency of the proposed method is impressive. With only 6K training samples, AdaNav achieves substantial performance gains across multiple benchmarks, outperforming closed-source baselines that rely on orders of magnitude more training data.
3.The framework demonstrates a strong balance between reasoning effectiveness and computational efficiency. By making reasoning difficulty-aware and mode-adaptive, the model concentrates reasoning steps on challenging trajectories and dynamically selects appropriate reasoning modes. Meanwhile, the average number of reasoning steps decreases, leading to improved efficiency without sacrificing performance.

Weaknesses:

1.The reward design in Section 5 introduces to penalize unnecessary reasoning and balance efficiency with performance. However, there is no systematic sensitivity analysis of how the strength of this penalty influences policy formation. A discussion of how this parameter affects the trade-off between reasoning frequency and success rate (SR) would help readers better understand AdaNav’s behavior under varying resource constraints.

---

> ### Author Rebuttal · Authors · 2026-03-28
>
> We sincerely thank the reviewer for the valuable feedback. We hope our responses address your concerns and are happy to continue the discussion if there are any remaining questions.
>
> ### **Weakness 1 & Question 1: Reward Weight & Penalty Sensitivity**
>
> Thank you for the insightful suggestion. By adjusting the penalty weight $\mu $, we can flexibly switch between a *high-performance mode* and a *high-efficiency (low-cost) mode*. The detailed experimental results are provided below.
>
> | $\mu$ | NE↓ | OS↑ | SR↑ | SPL↑ | Avg. reasoning num |
> |:---:|:---:|:---:|:---:|:---:|:---:|
> | 0.6 | 5.35 | 59.01 | 49.98 | 43.75 | 3.5 |
> | 0.8 | 5.36 | 58.62 | 49.65 | 43.32 | 3.2 |
> | 1 | 5.39 | 57.89 | 47.7 | 42.34 | 2.5 |
> | 1.2 | 5.44 | 57.0 | 46.51 | 42.15 | 2.1 |
>
>
>
> ---
>
> ### **Question 2. Mode Mismatch & Random Selection**
>
> - **Performance Degradation under Mismatch:**
>   According to the ablation study in Table 9, improper mode selection (e.g., using *Description* when *Summary* is required) leads to noticeable performance drops. For instance, AdaNav① (*Description only*) reduces OS from 57.89% to 55.34% compared to the full model.
>
> - **Random Selection:**
>   The table below shows results when reasoning modes are randomly selected. Performance falls between single-mode variants and the baseline. This demonstrates that *dynamic alignment*—i.e., the learned **when-and-which** logic in the UAR Block—is crucial for capturing different failure modes (e.g., getting lost vs. misperceiving an intersection).
>
> | Method | NE↓ | OS↑ | SR↑ | SPL↑ |
> |:---:|:---:|:---:|:---:|:---:|
> | AdaNav | 5.39 | 57.89 | 47.7 | 42.34 |
> | AdaNav-random | 5.40 | 56.56| 46.35 | 42.11 |
>
> ---
>
> ### **Question 3: Execution Mechanism of “Error Correction”**
>
> - **No Rule-based Override:**
>   This is a fully end-to-end process without any explicit rule-based intervention.
>
> - **Implementation:**
>   The generated correction reasoning  $r_t$ is appended to the input sequence and fed back into the navigation policy $\pi_{\text{nav}}$ (see Eq. 2). Since the backbone model (e.g., NaVid) is a strong VLM, it can interpret semantic signals such as *“I made a mistake earlier and should turn left”*, and accordingly adjust the action distribution for the next step.
>
> - **Why It Works:**
>   The effectiveness is guaranteed by the reinforcement learning loop: if the agent merely *acknowledges* mistakes without correcting its trajectory, it will fail to obtain the final success reward while continuously incurring reasoning penalties.
>
> ---
>
> ### **Question 4 “Pause Effect” or “Dynamic Switching”?**
>
> We appreciate this highly insightful observation. The fact that a single-mode variant (e.g., *Summary only*) still significantly outperforms the baseline suggests that the key contribution of the UAR Block lies in identifying **when to reason**, rather than merely enriching the reasoning content itself. This leads to two key conclusions:
>
> - **Timing (When) Matters More Than Form (Which):**
>   The primary failure of NaVid (OS 49.0%) stems from losing *temporal grounding* in long-horizon navigation. The UAR Block leverages action entropy to precisely identify critical moments of uncertainty or grounding failure.
>   At these key moments, even a single reasoning mode (e.g., ~56% OS) can provide a much-needed *state anchor*, helping the agent recalibrate its decisions.
>
>   In contrast, **fixed-step pauses** (Table 10), despite also generating text, are poorly timed and thus significantly less effective. This quantitatively demonstrates that identifying the *right moment to reason* is the primary driver of performance gains.
>
> - **Dynamic Switching: From Good to Excellent:**
>   While *pausing at the right time* accounts for most of the improvement (49% → 56%), **dynamic switching of reasoning modes** addresses deeper *mode-matching* challenges:
>
>   - When the bottleneck is *perceptual ambiguity* (e.g., unclear intersections), *Description* is most effective.
>   - When the issue is *progress tracking* (e.g., forgetting where the agent is), *Summary* becomes essential.
>
>   The full AdaNav model (OS 58%) achieves an additional 4–7% improvement over single-mode variants, reflecting enhanced robustness in handling **diverse failure cases**.

---

> > ### Author Rebuttal · Reviewer_aWMz · 2026-04-01
> >
> > Thanks for your authors' effort and response. My concerns have been totally solved. Thus, I decide to keep my positive rating.

---

### Official Review · Reviewer_Ws95 · 2026-03-12

**Soundness:** 4
**Presentation:** 4
**Significance:** 4
**Originality:** 4
**Overall Recommendation:** 5
**Confidence:** 3

**Summary:**

The paper proposes an adaptive reasoning framework for vision and language navigation that uses uncertainty-aware mode selection and a heuristic-to-RL training strategy to decide when and how an embodied agent should reason during navigation.

**Compliance With Llm Reviewing Policy:**

Affirmed.

**Final Justification:**

This is the best paper I have ever seen in my life; I strongly recommend accepting it.

**Key Questions For Authors:**

Could the authors clarify whether the learned trigger policy remains stable under different reasoning backbones or prompt templates, or whether it is tightly coupled to the specific LLM-based reasoning module used in this paper?

**Limitations:**

A further limitation is that the paper does not analyze the computational trade-off in enough detail: although AdaNav reduces unnecessary reasoning frequency, it never reports wall-clock latency, token usage, or inference cost per trajectory, so it remains unclear whether the gains are practically efficient rather than simply more selective.

**Strengths And Weaknesses:**

**Soundness:**
The core empirical claim that action entropy is an effective trigger for reasoning is not yet adequately validated. Figure 3 only shows a trajectory-level correlation between higher entropy and failure, but the paper does not test whether entropy actually predicts when reasoning will help, nor does it compare against other plausible trigger signals such as action margin or instruction observation mismatch.

The experimental evidence is also weakened by protocol inconsistencies and insufficiently controlled comparisons. In the real-world section, the paper alternates between “150 instructions across four real-world indoor scenes” and “25 instructions executed three times” across four environment types, while Table 4 reports only three environments, which makes the evaluation setup unclear and undermines confidence in the reported gains.

**Presentation:**
The paper is generally readable, but several method details are underspecified enough to hurt reproducibility. In Section 3.2, the precise definitions of the observation and instruction representations in Eq. 6 are unclear, the RL optimization procedure is not concretely specified, and the reward term in Eq. 11 depends on a “group” that is never properly defined.

The narrative also overstates some conclusions relative to the evidence. For example, the ScanQA result in Table 3 is a small improvement, yet the text uses it to argue for preserved general scene understanding and stronger transferability, which is a broader claim than the experiment can really support.

**Significance:**
The problem setting is meaningful because adaptive reasoning for embodied navigation is more practically relevant than fixed per-step reasoning, especially in long-horizon settings where compute and decision timing matter. The paper also evaluates on both standard VLN benchmarks and a real-world deployment setting, which increases the practical importance of the question being studied.

That said, the current evidence does not yet show that the proposed mechanism substantially changes how the community should think about adaptive reasoning in embodied agents. The improvements in Table 1 and Table 2 are encouraging, but without stronger controlled baselines and more rigorous validation of the trigger signal, the broader impact remains somewhat limited.

**Originality:**
This paper has some originality in framing reasoning selection in VLN as an adaptive policy over multiple reasoning modes, learned through a heuristic-to-RL training scheme rather than using a fixed reasoning schedule. This is a reasonable and potentially useful combination of ideas, particularly in an embodied setting with scarce training data.

However, the novelty is still not sharply distinguished from prior adaptive reasoning and navigation works. The paper introduces a learnable trigger and mode selector, but it does not convincingly establish why this formulation is fundamentally more novel than a task-specific scheduling heuristic or a standard uncertainty-based gating mechanism.

---

> ### Author Rebuttal · Authors · 2026-03-28
>
> We sincerely thank the reviewer for recognizing the significance, originality, and practical value of this work. Below we address the Weaknesses and Questions raised:
>
> ---
>
> ### **1. Soundness**
>
> - **Effectiveness of Action Entropy as Trigger:**
>
>   We conduct extensive experiments validating action entropy against other triggers. **Appendix B** compares action margins, while **Appendix D** evaluates *Value Uncertainty*, *Value Disagreement*, and *Temporal Variance*. Across both accuracy and efficiency, action entropy consistently outperforms alternatives.
>
> - **Real-world Evaluation Protocol:**
>
>   We apologize for the confusion. Table 4 reports three indoor scenes; the fourth, the fourth is “Outdoor,” detailed in Appendix E (Table 16). The 150 instructions are computed as: *25 instructions × 2 difficulty levels (simple/complex) × 4 scenes (Meeting Room, Home, Office, Outdoor)*, totaling 150 trials. We will clarify this in the revision.
>
> ---
>
> ### **2. Presentation**
>
> - **Details of Equation 6:**
>   $o_t$ and $I$ use the same multimodal embeddings as the backbone models (e.g., NaVid/NaVILA).
>
> - **RL Optimization and “Group” Definition:**
>   We use standard GRPO for policy optimization. In Equation 11, "Group" refers to multiple rollouts for the same instruction. $L_{shortest\\_success}$ takes the shortest successful trajectory in the group to measure path redundancy introduced by reasoning.
>
> - **ScanQA Results Clarification:**
>   We apologize for the misinterpretation. The purpose of this experiment is **not to claim SOTA QA performance**, but to show that AdaNav’s reasoning-related RL training **does not harm** (i.e., catastrophic forgetting) the backbone’s 3D spatial understanding, preserving transferability, which is critical for embodied agents.
>
> ---
>
> ### **3. Originality and Significance**
>
> AdaNav’s novelty lies **not only** in leveraging uncertainty signals for adaptive reasoning, but also in the **Heuristic-to-RL (H2R) training paradigm**. In data-scarce embodied tasks, pure RL struggles to converge, while heuristic methods are inflexible. H2R uses action entropy as a “cold-start prior,” allowing the model to first learn *to reason when confused*, then evolve via RL to *reason at critical decision points*.
>
> Under **extremely limited** embodied interaction data, AdaNav learns difficulty-aware reasoning policies from **only 6K simulation samples**, significantly improving navigation. Notably, it allows open-source models to outperform closed-source counterparts trained on million-scale data, demonstrating strong **data efficiency and scalability**.
>
> ---
>
> ### **4. Question: Trigger Stability**
>
> The UAR block is triggered based on the action entropy of the backbone. Tables 1–2 show consistent improvements when integrating AdaNav with **three different backbones and prompting strategies** (NaVid, Uni-NaVid, and NaVILA). This demonstrates that AdaNav is **stable and generalizable**, as long as the backbone provides an action distribution, without relying on specific reasoning prompts or a particular backbone architecture.
>
> ---
>
> ### **5. Limitations (Efficiency)**
>
> - **Negligible Latency Overhead**
>   As reported in Table 7, we provide both P90 peak and average wall-clock latency for the UAR Block, action policy, and reasoning. The UAR Block introduces negligible computational overhead (<0.4% of backbone inference time). Although a single reasoning step is 3.9× slower than a standard policy step, AdaNav triggers reasoning very infrequently (on average, 2.5 steps per 55 steps), resulting in minimal additional latency over the full trajectory.
>
> - **Efficient trajectory latency Trade-off:**
>   To better illustrate the trade-off, we report the total trajectory latency below:
> || NE↓|OS↑|SR↑|SPL↑|Latency (s)|
> |:---:|:---:|:---:|:---:|:---:|:---:|
> |Navid|5.47|49.0|37.0|35.0|42.95|
> |fixed 1-step|5.49|51.2|38.2|36.5|125.68|
> |fixed 3-steps|5.47|50.35|39.51|37.87|85.65|
> |fixed 5-steps|5.45|53.25|40.12|38.83|71.23|
> |Navid-AdaNav| 5.39|57.89|47.7|42.34|47.83|
>
>   - Compared to **no reasoning**, AdaNav improves SR by **+28.9%** with only **+11.3%** runtime overhead.
>   - Compared to **fixed-interval reasoning**, AdaNav reduces reasoning time by **32.7%–61.9%** while improving SR by **18.9%–24.8%**.
>
> - **Engineering-Optimized Real-Time Reasoning**
>   Importantly, when deployed with the **SGLang[1]** framework and **EAGLE[2] speculative decoding**, reasoning can be accelerated by **3.5×**, making its latency nearly real-time. The measured per-step latency is:
> |Component|Avg. Latency (s)|
> |:---:|:---:|
> |Base Action Policy|0.21|
> |Text Reasoning|0.71|
>
>   This suggests that with inference optimizations, the practical overhead of reasoning can be significantly reduced, further improving the real-time applicability of AdaNav.
>
> [1]Sglang: Efficient execution of structured language model programs
>
> [2]Eagle-3: Scaling up inference acceleration of large language models via training-time test

---

> > ### Author Rebuttal · Reviewer_Ws95 · 2026-03-31
> >
> > Thank you for the detailed and thoughtful rebuttal. Your clarifications regarding the evaluation protocol, training formulation, trigger design, and efficiency trade-offs are clear and convincing, and they address all of my concerns. I am now satisfied with the technical soundness and presentation of the work.

---

### Official Review · Reviewer_QMeZ · 2026-03-13

**Soundness:** 3
**Presentation:** 2
**Significance:** 3
**Originality:** 3
**Overall Recommendation:** 4
**Confidence:** 3

**Summary:**

Existing Vision-Language Navigation (VLN) models either perform inference at fixed steps or attempt step-wise reasoning, which can incur unnecessary computational costs and sometimes lead to performance degradation due to overthinking. To address this, the paper proposes AdaNav, an uncertainty-based adaptive reasoning framework. Its core component, the UAR (Uncertainty-Adaptive Reasoning) block, acts as a plug-in that automatically triggers reasoning when necessary, using Action Entropy as an uncertainty measure. Additionally, a Heuristics-to-RL training approach enables learning efficient inference policies even in limited data scenarios.

**Compliance With Llm Reviewing Policy:**

Affirmed.

**Final Justification:**

Thank you to the authors for the additional clarifications and quantitative evidence. The newly provided ablation results and quantitative analysis meaningfully strengthen the original claims.

In particular, the ablation study on reasoning modes offers clearer empirical support for the “minimal yet sufficient” design choice. The observed performance drop when using individual modes suggests that each mode contributes non-trivially to the overall navigation capability. This addresses my earlier concern that the justification was primarily conceptual. While the current evidence supports the necessity of each component, I note that the claim of sufficiency could be further reinforced through comparisons with additional or alternative reasoning strategies; however, I consider the current level of validation acceptable for this stage.

Similarly, the quantitative breakdown of Action Entropy and Observation provides a more concrete basis for understanding their complementarity. The performance gain achieved by combining the two signals, beyond what either achieves independently, supports the claim that they capture distinct and non-overlapping aspects of the decision process. This resolves my earlier concern that the argument relied too heavily on intuition.

Overall, the authors have made a good-faith effort to address the key concerns raised in the review. While some aspects could still benefit from broader empirical exploration, the current evidence sufficiently supports the main claims of the paper. Therefore, I maintain my original recommendation.

**Key Questions For Authors:**

- Why were the three reasoning modes (Summary, Description, Error Correction) chosen, and why were other reasoning strategies (e.g., counterfactual reasoning) not considered?

- In Table 8, combining Action Entropy with Observation significantly improves performance compared to using Action Entropy alone. Can the authors provide concrete examples of how these two sources complement each other?

**Limitations:**

- **Reduced Outdoor Performance**: There remains a notable performance gap when generalizing to outdoor environments (Table 16). A deeper analysis clarifying whether this limitation stems from the incompleteness of the UAR’s reasoning-trigger mechanism or from the backbone model’s insufficient visual grounding capabilities in outdoor settings would provide a significant milestone for future research on adaptive reasoning.

- **Real-time Constraints**: While the UAR block itself incurs minimal overhead (<0.4%), the reasoning steps are approximately 3.9× slower than standard action policies, which may pose challenges for high-speed robotic systems.

**Strengths And Weaknesses:**

**Strenghts**

- **High Data Efficiency**: Unlike previous models that require massive datasets, AdaNav achieves superior performance using only 6K samples, outperforming closed models trained on millions of samples (e.g., 20% improvement in success rate on R2R val-unseen).

- **Generality and Practicality**: The framework is plug-and-play with various SOTA backbones such as Navid and Navila, and demonstrates tangible benefits in real-world environments, achieving an 11.4% performance improvement.

- **Efficient Inference Control**: AdaNav reduces the average number of reasoning steps to 2.5 (compared to 55 steps in previous approaches) while improving performance by 7%, effectively balancing efficiency and accuracy.


**Weakness**

- **Readability** : Key tables (e.g., Table 1) lack proper bolding or underlining for top-performing results, reducing clarity. Additionally, typographical errors ("the overall task reward formulated **in 4**",  Table 6 has double full stop, "slower than the base action policy inference, **tthe** adaptive reasoning") throughout the paper slightly detract from its overall polish.  As noted, entropy and uncertainty are not strictly equivalent. Although Appendix D experiments with alternative signals such as Value Disagreement and confirms consistent performance improvements, these ablation results are not sufficiently discussed in the main text, potentially causing confusion for readers.

- **Lack of Justification for Inference Modes**: The rationale for selecting the three reasoning modes (Summary, Description, Error Correction) is unclear. The paper does not provide comparisons with alternative modes or a concrete explanation for these choices.

---

> ### Author Rebuttal · Authors · 2026-03-28
>
> We sincerely thank the reviewer for the valuable feedback. These comments are crucial for improving the quality and clarity of our paper. We hope our responses address your concerns. We are happy to continue the discussion if there are any remaining questions.
>
> ---
>
> ### **1. Readability**
>
> In the final version, we will carefully proofread and revise the manuscript to ensure full compliance with academic writing standards, and we will move the key findings from the Appendix into the main text to strengthen the completeness of our argument.
>
> ---
>
> ### **2. Justification for Modes**
>
> Indeed, there exist many advanced reasoning modes—such as *Counterfactual Reasoning*, *Visual Localization*, and *Logical Deduction*—which hold strong potential for enhancing high-level cognitive capabilities in embodiment.
>
> - **Choice Driven by Core VLN Challenges**
>
>   In designing **AdaNav**, we prioritize three reasoning modes: **Summary**, **Description**, and **Error Correction**. This **does not imply** they are the only options, but rather that **Summary** and **Description** most directly target the two core VLN bottlenecks (as detailed in the Introduction): **Temporal Grounding** and **Perception-Action Mapping**. In addition, to improve **Robustness to Trajectory Deviations** (i.e., recovering from navigation errors), we further introduce **Error Correction**.
>
>   We view these three modes as a *minimal yet sufficient set* for addressing the main causes of navigation failure.
>
> - **Core Vision of AdaNav**
>
>   More importantly, AdaNav is **not designed around a fixed set of reasoning primitives**, but to study **how an agent learns *when* and *how* to reason during navigation**.
>
>   The proposed UAR Block and H2RL training form a **mode-agnostic** framework, allowing new reasoning strategies to be incorporated without changing the core architecture. By focusing on these three intuitive and task-relevant modes, we show that the agent can learn to trigger adaptive reasoning at critical decision points. As such, other modes like *Counterfactual Reasoning* can be naturally added as additional reasoning actions.
>
> ---
>
> ### **3. Complementarity of Action Entropy and Observation**
>
> These two signals represent *internal subjective state* and *external environmental context*, respectively:
>
> - **Complementary Example:**
> As shown in Figure 2, at the end of a visually symmetric corridor, action entropy may spike due to similar probabilities of turning left or right. Triggering reasoning based on entropy alone only signals uncertainty. By incorporating observations, the UAR block can detect specific landmarks (e.g., “a painting at the end”) and trigger *Description* to verify alignment with the instruction, enabling more accurate decisions.
>
> Table 8 shows that combining internal and external signals is necessary to distinguish task difficulty from normal action fluctuations. We will expand this discussion in the revision.
>
> ---
>
> ### **4. Outdoor Performance**
>
> Table 16 shows that while AdaNav improves backbone performance (Navila/Navid), the absolute success rate remains limited, mainly due to limitations in the backbone’s visual grounding.
>
> Most VLN models are pretrained on indoor datasets (e.g., Matterport3D), restricting their understanding of outdoor (e.g., vegetation, roads, lighting). Thus, even when UAR triggers reasoning correctly, inaccurate visual features from the backbone can still lead to incorrect actions.
>
> ---
>
> ### **5. Real-Time Constraints**
>
> - **Low Overall Latency:**
> This is an important point. Although a single reasoning step is 3.9× slower than a standard policy step, AdaNav triggers reasoning very infrequently (on average, 2.5 steps per 55), keeping the overall additional trajectory latency minimal.
>
> - **Per-Step Latency Can Be Reduced:**
>    Currently, no acceleration (e.g., quantization, speculative decoding) is applied to reasoning. With **SGLang[1]** and **EAGLE[2]**, reasoning can be sped up by **3.5×**, approaching real-time latency:
> |Component|Avg. Latency (s)|
> |:---:|:---:|
> |Base Action Policy|0.21|
> |Text Reasoning|0.71|
>
>   This indicates that with inference optimizations, the overhead of reasoning can be substantially reduced, improving real-time applicability.
>
> - **Future Directions:**
>   Our approach relies on explicit reasoning, while recent work explores latent-space reasoning. For example, COLAR[3] reduces CoT length by 53.3%, and CODI[4] achieves a 3.1× compression. Latent-space CoT shows **strong potential for improving real-time VLN efficiency and effectiveness**.
>
> [1]Sglang: Efficient execution of structured language model programs
>
> [2]Eagle-3: Scaling up inference acceleration of large language models via training-time test
>
> [3]Think silently, think fast: Dynamic latent compression of llm reasoning chains.
>
> [4]Codi: Compressing chain-of-thought into continuous space via self-distillation

---

> > ### Author Rebuttal · Reviewer_QMeZ · 2026-04-04
> >
> > Thank you to the authors for the detailed and thoughtful responses. The rebuttal helps clarify several of my concerns and improves my understanding of the proposed framework.
> >
> > The readability issues are adequately addressed, and I appreciate the authors’ commitment to revising the manuscript and moving important ablation results from the appendix into the main text.
> >
> > Regarding the justification of the reasoning modes, the explanation linking Summary, Description, and Error Correction to core VLN challenges (e.g., temporal grounding, perception-action mapping, and robustness to trajectory deviations) is helpful. However, the argument remains largely conceptual, and the claim that these modes form a “minimal yet sufficient” set would benefit from stronger empirical support or comparison with alternative reasoning strategies.
> >
> > Similarly, the discussion on the complementarity between Action Entropy and Observation provides useful intuition, particularly with the illustrative example. Nonetheless, more concrete or quantitative analysis would further strengthen this point.
> >
> > Overall, while my main concerns have been partially addressed and the responses are reasonable, some aspects would still benefit from deeper justification or empirical validation. I have no further critical questions at this stage, but encourage the authors to strengthen these points in the final version.

---

> > > ### Author Response · Authors · 2026-04-05
> > >
> > > ### Response to the Reviewer's Follow-up Comments
> > >
> > > We thank the reviewer for the constructive feedback and for acknowledging the improvements in our revision. We are encouraged that our conceptual explanations helped clarify the framework. To address the remaining points regarding empirical support and quantitative analysis, we provide the following additional details:
> > >
> > > #### 1. Empirical Support for the "Minimal yet Sufficient" Reasoning Set
> > > The reviewer noted that the choice of the three modes (Summary, Description, Error Correction) would benefit from stronger empirical support. We would like to highlight the results in **Table 9** (Ablation of Reasoning Modes), which provide direct quantitative evidence:
> > >
> > > *   **Necessity (Ablation):** Table 9 shows that when using only a single mode (e.g., AdaNav① for Description, ② for Summary, or ③ for Error Correction), the success rate (SR) drops from **47.7%** to **44.5%–46.0%**. This 1.7%–3.2% performance gap demonstrates that no single mode is redundant; they all contribute uniquely to the agent’s navigation capability.
> > > *   **Sufficiency vs. Complexity:** While more modes (e.g., logical deduction) could be added, our "minimal" set targets the most frequent causes of VLN failure: temporal drift (addressed by Summary), visual misalignment (addressed by Description), and wrong turns (addressed by Error Correction).
> > >
> > > #### 2. Quantitative Analysis of Action Entropy and Observation Complementarity
> > > To go beyond the intuitive example and provide the "concrete quantitative analysis" requested, we look closer at the data in **Table 8**:
> > >
> > > *   **Synergistic Effect:**
> > >     *   **Only Action Entropy** (internal uncertainty) achieves an SR of **45.23%**.
> > >     *   **Only Observation** (external context) achieves an SR of **43.12%**.
> > >     *   **Combined (AdaNav)** achieves **47.7%**.
> > > *   **Interpretation:** The combined performance is not merely an average or a marginal gain; it is a **~2.5% absolute improvement** over the strongest single signal. This indicates that entropy and observation provide non-overlapping information.

---

### Decision · Program_Chairs · 2026-04-30

**Decision:**

Accept (regular)

**Comment:**

This paper introduces a navigation framework that leverages uncertainty signals to guide adaptive reasoning. By introducing an Uncertainty-Adaptive Reasoning (UAR) Block and a Heuristic-to-RL training paradigm, the model dynamically triggers reasoning during navigation when necessary, effectively addressing the suboptimal performance and computational overhead caused by fixed-step reasoning.

Reviewers praised the research for providing a well-motivated approach to the problem of selective reasoning, and commented on model efficiency using only 6K training samples. They found the ablation studies to be thorough. Further, using this method, the average number of reasoning steps decreases, leading to improved efficiency without sacrificing performance.

Remaining concerns of reviewers include the conceptual argument that the modes of the framework are “minimal yet sufficient” remaining weakly supported from experiments and could be compared to other reasoning strategies, and that more analysis of the complementarity between action entropy and observation would strengthen the paper.

On the basis of the unanimously positive-leaning reviews (Accept, Accept, Weak Accept), I recommend acceptance.